# Prescribing Exercise to Cancer Patients Suffering from Increased Bone Fracture Risk Due to Metastatic Bone Disease or Multiple Myeloma in Austria—An Inter- and Multidisciplinary Evaluation Measure

**DOI:** 10.3390/cancers15041245

**Published:** 2023-02-15

**Authors:** Richard Crevenna, Timothy Hasenoehrl, Christoph Wiltschke, Franz Kainberger, Mohammad Keilani

**Affiliations:** 1Department of Physical Medicine, Rehabilitation and Occupational Medicine, Medical University of Vienna, 1090 Vienna, Austria; 2Department of Radiology and Osteology, Medical University of Vienna, 1090 Vienna, Austria

**Keywords:** assessment, risk, fractures, pathologic, rehabilitation

## Abstract

**Simple Summary:**

Some cancer patients are at increased risk of fractures due to their disease. Since these patients would still benefit from physical exercise overall, they should be given exercise recommendations that take their individual musculoskeletal situation into account. This article describes how this process should be handled according to Austrian experts. Each patient should be considered an individual case managed by his or her medical case manager (physiatrist/rehabilitation specialist). There should be specialists who assess the fracture risk (radiologist, oncologist, orthopedist, and radiation specialist), specialists who assess cardiovascular risk (internist and cardiologist), specialists who choose suitable exercises (sport scientist and physiatrist), and specialists for additional physical treatment (physiatrist and physical therapist).

**Abstract:**

Introduction: In the current absence of specific functional fracture risk assessment technology, the planning of physical exercise interventions for cancer patients suffering from increased bone fracture risk remains a serious clinical challenge. Until a reliable, solely technical solution is available for the clinician, fracture risk assessment remains an inter- and multidisciplinary decision to be made by various medical experts. The aim of this short paper is depicting how this challenge should be approached in the clinical reality according to Austrian experts in cancer rehabilitation, presenting the best-practice model in Austria. Following referral from the specialist responsible for the primary cancer treatment (oncologist, surgeon, etc.), the physiatrist takes on the role of rehabilitation case manager for each individual patient. Fracture risk assessment is then undertaken by specialists in radiology, orthopedics, oncology, and radiation therapy, with the result that the affected bone regions are classified as being at highly/slightly/not increased fracture risk. Following internal clearance, exercise planning is undertaken by a specialist in exercise therapy together with the physiatrist based on the individual’s fracture risk assessment. In the case in which the patient shows exercise limitations due to additional musculoskeletal impairments, adjuvant physical modalities such as physiotherapy should be prescribed to increase exercisability. Conclusion: Exercise prescription for cancer patients suffering from increased fracture risk is an inter- and multidisciplinary team decision for each individual patient.

## 1. Introduction

Evidence of the benefits of physical exercise in cancer patients has grown exponentially over the past two decades. Physical exercise benefits cancer patients by preserving and increasing physical function and the health-related quality of life, and via systemic anti-inflammatory and immune system-modulating effects [1,2,3]. In some cancer entities, physical exercise has even become the treatment for a specific treatment-related side-effect. In prostate cancer patients under androgen deprivation therapy, for example, muscle loss can be successfully treated with resistance exercise [3,4]. In breast cancer patients suffering from breast cancer-related lymphedema, resistance exercise can be performed safely and is beneficial for arm function and even lymphedema levels [5,6]; in cancer generally, cancer-related fatigue can be successfully treated with physical exercise [7].

These advances in cancer rehabilitation research would have been impossible if early research had not challenged some of the then long-established paradigms and demonstrated, in early frontier research, that exercise not only is not harmful but can even be beneficial in cancer patients. Some of the first publications worldwide in this regard were published at Department of Physical Medicine, Rehabilitation and Occupational Medicine of Medical University of Vienna, Austria, showing then groundbreaking and mind-opening results in a breast cancer patient suffering from inflammatory recurrence [8], a patient suffering from advanced hepatocellular cancer with brain metastases [9], and a patient suffering from advanced breast cancer with metastatic bone disease [10] during a time when cancer patients were still advised to be inactive and rest. These very early publications laid the foundation for the establishment of exercise oncology as an important field of research even against what was then common knowledge.

Meanwhile, physical exercise also plays a pivotal role in the relatively young field of “prehabilitation” for cancer patients. Several benefits of exercise in cancer prehabilitation have been described, such as the reduction in post-treatment (for example, postoperative) complications, the shortening in inpatient stay, the improvement in survival, the reduction in relapse rate, and the improvement in mental health [11,12,13,14,15,16].

The unanimous tenor in sports oncology research has led to clear statements and recommendations in favor of increasing exercise levels in cancer patients from the world’s most renowned sports research societies, such as the American College of Sports Medicine (ACSM) [17,18], the Australian Association for Exercise and Sport Science [19], and the International Society of Exercise and Immunology [20]. Even the non-exercise-specific American Cancer Society has emphasized the importance of physical activity in cancer patients and has published guidelines for their clinicians on how to best approach physical activity in cancer patients [21]. In clinical practice, these recommendations—to assess, to advice, and to refer—are more difficult to follow in some cancer entities than in others. Particularly, in cancer types that affect bone health, fracture risk might be increased at the affected skeletal site. This can be due to either being directly affected by the cancer itself, such as in multiple myeloma [22] or in cancer entities that are more likely to develop metastatic bone disease (e.g., prostate, breast, lung, and renal cancer) [23], or generally by treatment side-effects, such as chemotherapy or radiation damage [24].

Bone lesions can notably increase morbidity and mortality in cancer patients [25]; therefore, physical exercise, which naturally includes the mechanical loading of the musculoskeletal system, needs to be treated as a potential risk factor. The practical problem we are now facing is that while oncological and hemato-oncological therapies improve survival rates in patients with metastatic bone disease or multiple myeloma [26,27,28,29], the cancer- and/or treatment-related side-effects, which in other cancers are successfully treated with supportive exercise therapy, cannot be similarly treated without assessing fracture risk first. However, like patients suffering from other cancer entities, patients suffering from metastatic bone disease or multiple myeloma should also be provided with exercise recommendations, as they have, in several cases, a good prognosis. Moreover, considering the significance of preserving physical function and increasing the quality of life, as well as the benefits of systemic myokine activity, cancer patients with locally increased fracture risk should also exercise. Bone regions with increased fracture risk need special assessment and exercise handling. A challenge in this context is the assessment of each patient’s individual fracture risk.

In this regard, there are currently several different approaches in use. On the one hand, there are radiologic scores, such as Mirel’s score [30], Spine Instability Neoplastic Score (SINS) [31], or Myeloma Spine and Bone Damage Score (MSBDS) [32], which make the radiologic assessment of bone stability possible. On the other hand, there is the practical method of excluding skeletal sites affected by bone lesions by employing any exercise-induced mechanical stress [33].

Since, to date, there is currently no single method for assessing bone fracture risk as a basis for exercise recommendations, a multi-professional, interdisciplinary team decision seems to be the optimum choice in clinical practice. This has been supported by recent recommendations by “Campbell et al., 2022” [34] and “Hart et al., 2022” [35], who discussed the real impact of an issue for which there is not yet a standard procedure. The aim of this article is depicting the current inter- and multidisciplinary exercise prescription procedure for these patients in Austria.

## 2. Best-Practice Model for Exercise Prescription in Austria

In Austria, exercise prescription for cancer patients at increased fracture risk is treated as an inter- and multidisciplinary approach. History taking and clinical examination (radiographic findings and bone scans), certain laboratory parameters, resting electrocardiogram (ECG), echocardiography findings, muscular strength testing, and (spiro)ergometry are required for planning individual programs. Experts in different disciplines (physiatrist, oncologist, radiologist, radiation therapist, laboratory physician, sports scientist, nutritionist, physiotherapist, etc.) are involved in this process.

The key point is to perform sufficient fracture risk assessment, as well as assessment of other cancer complications and of co-morbidities (such as cardiovascular diseases). Besides the skeletal considerations, in multiple myeloma, there are special clinical features and contraindications, such as hypercalcemia and monoclonal gammopathy [36], which have to be taken into account before as well as during the process of exercise prescription and conduction.

The initiative to pursue exercise prescription needs to come from the specialist who is responsible for the primary cancer treatment (oncologist, surgeon, etc.), who—after careful consideration of the risk–benefit ratio, including cardiovascular risk, as well as the patient’s prognosis—consults the physiatrist. The specialist in physical and rehabilitation medicine then takes on the role of case manager, as he or she is the one who will ultimately prescribe exercise therapy. Together, they contact the radiologist. The radiologist evaluates the fracture risk using imaging methods and first decides if the obtained imaging is sufficient or if further imaging needs to be undertaken. Depending on the location—spine, long bone, or other bone—the radiologist uses established scores, such as Mirel’s score [30] or SINS [31], for establishing the diagnosis. Since none of the established fracture risk scores have been developed for the evaluation of fracture risk during physical exercise nor do they include all necessary factors, such as bone architecture and geometry or osteosarcopenia, the assessment is strongly influenced by clinical expertise. During this process, orthopedic and oncologic counseling regarding location, size, and type of bone lesion is highly recommended. The final outcome of this multidisciplinary fracture risk assessment can be one of the following diagnoses: the affected bone region can be loaded like a healthy structure (no increased risk); exercise-related physical loading should match and not exceed activities of daily living (slightly increased fracture risk); the affected region should evade any additional physical stress (highly increased fracture risk).

However, in our opinion, pathological fractures and spinal cord compression are to be excluded from exercise (Table 1).

Furthermore, there are several further contraindications that have to be considered during the whole procedure (Table 1).

In the next step, the sport scientist plans the exercise program together with the specialist in physical and rehabilitation medicine considering the following: first, the individual patient’s situation in regard to (cancer-related) deconditioning; second, the patient’s needs and time resources; third, the fracture risk assessment. Based on “Gottlob’s flux of force principle, 2020” [37], the exercise therapist assesses each potential exercise in regard to muscle activation. They, therefore, know beforehand which muscles are active as agonists, and respectively, synergists; which joints and bones are loaded dynamically, and respectively, statically; and how the resulting force vectors affect the skeleton. If during the practical exercise process, it is seen that the patient is unable to perform specific exercises due, e.g., to musculoskeletal problems, in an additional step, physiotherapy can be prescribed with the aim of increasing exercisability. Moreover, providing nutrition counseling could ensure full efficacy of the exercise program. The only medical specialist who is involved in all steps and who keeps track of the whole process is the specialist in physical and rehabilitation medicine. He or she can additionally evaluate and prescribe physical medicine modalities to treat further side-effects of cancer treatment.

Depending on the circumstances, this multidisciplinary process can either be conducted consecutively or, in an optimized clinical environment, via a rehabilitation tumor board [38,39].

## 3. Discussion and Conclusions

The above-noted best-practice model for exercise prescription is based on the decade-long practical experience of the involved specialists [8,9,10,39,40,41,42]. It shows the importance of inter- and multidisciplinary cooperation and proves that each specialty has its own specific expertise and relevance for optimized patient care. This clinical experience is of particularly high relevance, because in contrast to the fracture risk evaluation in osteoporotic bone, fracture risk assessment in metastatic bone is underexplored. In osteoporosis, measuring bone mineral density (BMD) via dual-energy X-ray absorptiometry (DEXA) has been established as the diagnostic marker for decades [43]. A T-score with ≤ −2.5 standard deviation below the BMD of a younger, healthy population of the same sex has been defined as a clinically relevant indicator of osteoporosis [43]. However, as BMD alone does not adequately predict fracture risk [44], various approaches are used to specify fracture risk in osteoporosis. One approach is utilizing fracture risk assessment tools. The globally most widely used tool is the FRAX^®^ 10-year fracture probability model, which incorporates factors such as age, sex, bodyweight, history of fractures, smoking or alcohol consumption, and others [45]. It is, however, criticized, because it does not include fall history [46]. The second widely used score is the QFracture score, which incorporates more than 20 risk factors, including falls [47]. It is, however, criticized for being very time-consuming [46]. Besides theses scores, as another approach to specify fracture risk, Trabecular Bone Score (TBS) can be used as additional radiologic assessment for the evaluation of the bone microarchitecture [44]. In addition to the previously mentioned parameters, the history of falls, immigration status, and type 2 diabetes mellitus have been suggested [48]. A lot of literature studies about fracture risk assessment in osteoporotic bone tissue exist. However, this knowledge cannot be directly adopted in malignant bone disease, as these pathologic fractures do not follow the usual patterns seen in patients with osteoporosis [49]. To address this challenge in malignant bone disease, several scores are used in clinical practice. The first published and still widely used score is Mirel’s score [30]. It incorporates the localization, pain, type, and size of the metastasis and defines a cut-off for increased fracture risk. However, Mirel’s score has been developed for long bones only with the aim to predict the necessity for prophylactic fixation surgery. It does not provide specific information regarding the load ability of the affected bone, which would be the key information needed for exercise therapy. Another scoring system is Spine Instability Neoplastic Score (SINS), by “Fisher et al., 2010” [31]. SINS assists in assessing the fracture risk in spinal bone and incorporates location, pain, type of lesion, spinal alignment, and vertebral body collapse, as well as the posterolateral involvement of the spinal elements, in its model [31]. However, SINS has been developed for the identification of fracture risk in spinal bone for surgical referral. It heavily relies on radiographic diagnostics and does not account for previous medical treatment, such as radiation therapy,; the extent of diffuse bone involvement; or poor bone quality [50]. Another scoring tool is MSDBS, which has been recently specifically developed for bone lesions in multiple myeloma patients [32]. MSDBS focuses on the location and size of the bone lesion. Its advantages are that it can be used in CT images, which are more easily available than MR images, and that it is fast and easily reproducible [32]. However, similar to all other common tools, it has not been developed with the aim of assessing functional bone stability for exercise recommendation purposes.

In regard to increasing the precision of fracture risk assessment, besides these scoring tools, technical innovations such as patient-specific finite element models seem to have potential [51,52]. While “Benca et al., 2019” tested their fracture prediction model in vitro [51], “Eggermont et al., 2020” [52] tested their fracture prediction model in vivo. They compared their finite element model with established clinical assessment methods. Depending on the size of the axial cortical involvement of the metastasis in the femur, they followed established clinical recommendations and defined lesions over 30 mm as being at high risk and those below 30 mm as being at low risk of fracture [52,53]. This cut-off corresponded with their finite element model’s critical failure load of 7.5 times the bodyweight. Their finite element model achieved a negative predictive value of 100% and a positive predictive value of 39% in their patient population, with both values being superior to the clinical guidelines (95% and 19%) [52]. As promising as these results are, there are still some limitations to broad clinical implementation. First, the fracture risk models of Benca et al. [51] and “Eggermont et al., 2020” [52] are limited to metastases in the femur, and second, the effects of muscular traction were not evaluated [51]. Moreover, in clinical practice, the high radiation doses necessary during quantitative computed tomography are problematic, particularly in patients who undergo frequent quantitative computed tomography scanning due to rapid growth of bone metastases.

Nevertheless, despite these numerous different fracture risk assessment approaches and besides the lack of an assessment gold standard for exercise prescription purposes, there is one key diagnostic problem. We still do not know enough about real-life vs. exercise-related fracture risk in patients suffering from bone lesions. Although it is common in the development of specific assessment methods to compare predicted versus real fractures in a patient population at risk [30,52], to our knowledge, nobody has evaluated, to date, in which situations malignant bones fracture. Does exercising in a potentially “safe exercise environment” need to be considered a fracture risk increase? Or is this not the case, because metastatic bones predominantly fracture during ADLs or falls and not during exercise training? Do we implement objective fall risk assessment sufficiently or is this generally overestimated in clinical practice [54]?

These are the question that have yet to be resolved. As this might be impossible and especially overly time-consuming for a single center, a centralized database is needed where metastatic, especially myeloma-related, bone fractures are collected regarding their etiology. Only then, it will be possible to assess patients’ “true” fracture risk with regard to exercise. In the meantime, considering the strengths and limitations of the available tools, exercise-related fracture risk assessment, along with practical implementation in patients, remains a process that should currently still rely to a large extent on the clinical expertise of a multi- and interdisciplinary team.

Considering this research gap, we would still like to hypothesize from our clinical experience that bone fractures rarely occur during medical exercise but rather during activities of daily living, when there is no specific focus on careful spine handling or fall risk. Considering the prescription process depicted above, we are convinced that medical exercise is a safe environment. However, there is a need for a multi-centric database for the etiology of metastatic bone lesion fractures.

## Figures and Tables

**Table 1 cancers-15-01245-t001:** Assessment and contraindications for exercise in patients suffering from metastatic bone disease or multiple myeloma.

Inter- and Multidisciplinary Approach
Evaluation of fracture risk and evaluation of other cancer complications (e.g., hypercalcemia and monoclonal gammopathy in multiple myeloma) and of co-morbidities (such as cardiovascular diseases). Patient history and clinical examination—radiographic findings and bone scans. Laboratory parameters. Resting electrocardiogram. Echocardiography findings. Muscular strength testing. (Spiro)ergometry.
**Contraindications**
Untreated unstable osseous lesions (especially pathological fractures and spinal cord compression) are excepted from physical loading during exercise (highly increased fracture risk in these regions). Uncontrolled elevated blood pressure. Acute myocardial infarction. Unstable angina pectoris. Decompensated heart failure. Uncontrolled arrhythmia. Third-degree heart block. Aortic aneurysm. Aortal stenosis. Dyspnea. Chest pain. Uncontrolled metabolic disease. Uncontrolled epilepsy. Acute systemic diseases and exacerbations. Acute infections. Fever. Significant decline in cognitive performance. Hemoglobin level < 8/dL. Thrombopenia < 20 × 10^9^/L. Untreated hypercalcemia, bone marrow aplasia, and insufficient renal function (in addition to multiple myeloma).

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
