# Peer review of "Prescribing Exercise to Cancer Patients Suffering from Increased Bone Fracture Risk Due to Metastatic Bone Disease or Multiple Myeloma in Austria—An Inter- and Multidisciplinary Evaluation Measure"

_cancers, 2023, doi:10.3390/cancers15041245_

Round 1
Reviewer 1 Report
1. Are there controversies in this field? What are the most recent and important achievements in the field? In my opinion, answers to these questions should be emphasized. Perhaps, in some cases, novelty of the recent achievements should be highlighted by indicating the year of publication in the text of the manuscript.
2. The results and discussion section is very weak and no emphasis is given on the discussion of the results like why certain effects are coming in to existence and what could be the possible reason behind them?
3. Results and conclusion: The section devoted to the explanation of the results suffers from the same problems revealed so far. Your storyline in the results section (and conclusion) is hard to follow. Moreover, the conclusions reached are really far from what one can infer from the empirical results.
4. The discussion should be rather organized around arguments avoiding simply describing details without providing much meaning. A real discussion should also link the findings of the study to theory and/or literature.
5. English is modest. Therefore, the authors need to improve their writing style. In addition, the whole manuscript needs to be checked by native English speakers.
Author Response
Dear Reviewer 1,
Thank you for taking time to review our manuscript.
After carefully reading your comments and suggestions, we as authors of this manuscript must collectively state that we cannot reconcile them with our manuscript. Your comments 2, 3, and 4 explicitly refer to a (low quality) intervention study. We however, submitted a viewpoint article so your points of criticism just don’t make sense in this context. And also comments 1 and 5 do not really fit.
We therefore understand that there was an issue, possibly of a technical nature, during the peer review process that resulted in the wrong peer review report being sent to us.
Might this have been the case?
Please read our detailed comments immediately afterwards as to why the current points of criticism could not be reconciled with our manuscript.
- Are there controversies in this field? What are the most recent and important achievements in the field? In my opinion, answers to these questions should be emphasized. Perhaps, in some cases, novelty of the recent achievements should be highlighted by indicating the year of publication in the text of the manuscript.
Thank you for your remark. Unfortunately, we are unable to integrate its full content into our manuscript. In the field of exercise oncology is a consensus that any cancer patient should be enabled to participate in exercise training. The challenges of fracture risk assessment in a specific population of cancer patients are well known. There is no 'gold standard' how to approach this challenge and since it is fundamentally an assessment problem that cannot be easily solved in the near future by e.g. a technical solution, various practical strategies exist. These strategies are all very similar at their core, so there is no major controversy in this area. Sharing of these strategies should inspire other researchers in the same field, which is the aim of this viewpoint paper.
- The results and discussion section is very weak and no emphasis is given on the discussion of the results like why certain effects are coming in to existence and what could be the possible reason behind them?
Thank you for this remark. Unfortunately, we are unable to understand how this comment fits to our manuscript. Our manuscript comprises neither a results nor a discussion section. We submitted a viewpoint paper, where we depicted the “Austrian approach” to a clinical challenge to which there are no guidelines established yet. Therefore, there are no “results” or “effects”.
- Results and conclusion: The section devoted to the explanation of the results suffers from the same problems revealed so far. Your storyline in the results section (and conclusion) is hard to follow. Moreover, the conclusions reached are really far from what one can infer from the empirical results.
Thank you for this remark. Unfortunately, we are unable to understand how this comment fits to our manuscript. Our manuscript comprises neither a “section devoted to the explanation of the results” nor “empirical results”.
- The discussion should be rather organized around arguments avoiding simply describing details without providing much meaning. A real discussion should also link the findings of the study to theory and/or literature.
Thank you for this remark. Unfortunately, we are unable to understand how this comment fits to our manuscript, which is not an original study but a viewpoint article.
- English is modest. Therefore, the authors need to improve their writing style. In addition, the whole manuscript needs to be checked by native English speakers.
Thank you for this remark. Unfortunately, we cannot make sense of it. The authors of this manuscript together have published over 600 scientific articles listed in PubMed/Medline, numerous of them in top-ranked international journals, never with the need for a language revision. We are therefore quite confident that the language quality is more than adequate.
We discussed this issue with the editor and asked them to help solve it.
Reviewer 2 Report
This manuscript describes an inter- and mltidisciplinary approach to prescribe the most suitable physcal exercise program for patients with a fracture risk due to bone metastasis. Overall, this manuscript is well written and the topic is interesting. However, there is a lack of evidence for the proposed method, as well as a lack of discussion and consideration for more novel and promising methods for assessing fracture risk (taking into account bone biomechanics). Therefore, it is difficult to evaluate the benefits and efficacy of this best practice model, and it does not seem to be easily reproducible.
Major issues
1) The authors state that assessment of each patient's individual fracture risk is a challenge, but how they address this challenge is arguable. Although a multi-professional and interdisciplinary team decision seems indeed to be an optimum in clinical practice, their approach is mainly based on clinicians' experience and lacks scientific evidence. For exemple, the radiologist estimates the fracture risk using the Mirels' score, but this clinical score lacks specificity and is highly subjective. Other promising methods have been proposed in the literature, such as the use of patient-specific finite element models (Eggermont et al, Bone, 2019). This should be discussed and an estimation of the precision of their fracture risk assessment should be provided.
2) The link between fracture risk assessment and the diagnosis of how an affected bone region can be loaded merits further precision. Several factors affect metastatic bone strength, such as bone density and geometry, bone architecture, characteristics of the boney lesion... Therefore, it is unclear how a diagnosis can be done from lesion features alone. Evidence of its accuracy is missing.
3) How the exercise therapist adapts their knowledge of how resulting force vectors affect the skeleton in the case of metastatic bone?
4) 15 out of the 43 references are self-citations. A more thorough comparison of the proposed approach with the literature could be provided.
Author Response
Dear Reviewer,
Thank you for your constructive feedback. We revised major parts of the manuscript in this regard and hope that it is now to your satisfaction. Please see our point-to-point answers for details.
Ad 1) Thank you very much for your valuable comment. We understand that you are referring to the sentence, “The radiologist then bases some of his diagnosis on established scores like for example the Mirel’s score” in your key point of criticism. First, we want to clear up, that the Mirel’s Score is not the only score used. Since it has been used widely, it was just the one example we chose.
Second, we are familiar with finite element models for the use of fracture risk assessment, particularly as one of the researchers of our university, who we cooperate with, has published specifically in this field (Benca E, e.g. PMID: 31311994). As promising these finite element models are, there are still some key limitations for their clinical implementation, besides the lack of wide availability. These models work rather well for the femoral bone however, they do not include muscle traction into their models. Aim of these fracture risk assessments are always recommendations for exercising, a physical activity which involves muscle traction, which can either increase or decrease mechanical loading of an effected bone region, depending on the posture and the force of flux and the associated muscle activity. The specific challenge we face now with these finite element models is, that not even in a rather simple muscular environment like the hip and leg muscles, muscle traction cannot be implemented into these models yet. Even less, this is possible for the spine due to the complexity of the trunk muscles, or the sternum, or the upper extremities. Therefore, finite element models will for sure play a pivotal role for future assessment techniques, at the time, they are just not developed enough to rely on them more or less than other techniques. This is why we think, that to date clinical experience is still a major factor for assessing exercise-related fracture risk.
Following your suggestions, we included a new paragraph specifically about this topic (page 5, line 224-241).
Ad 2) Thank you for this remark and for pointing out, that this information needed clarification. We addressed this topic by adding some information to the best practice model and by devoting a detailed paragraph regarding the fracture-risk assessment in osteoporosis and how it compares to the assessment in metastatic bone lesions. (page 4, line 184 into page 5, line 223)
Ad 3) Thank you for this comment. It helped us to realize that we should have included this theoretical input in the manuscript.
The basis for the exercise therapist’s knowledge is the “Flux of force principle” published by Axel Gottlob in Differenziertes Krafttraining mit Schwerpunkt Wirbelsäule, 2020, 5th Edition, Munich Jena, which is adapted for fracture risk specific purposes. The basic principle includes subsequent methodological questions which need to be answered to allow the estimation if a specific exercise exerts mechanical loading on a musculoskeletal structure.
- How does the flux of force go through the body?
The exercise therapist needs to answer the question of where the resistance enters the body and where the kinetic chain closes, which means, at which point the force is diverted into the floor or into the cushion of the exercise machine.
- Which joints and muscles are involved dynamically as well as statically?
Each joint and muscle which lies within the flux of force is either involved dynamically or statically.
- How are these joints/regions stabilized?
Each active muscle (chain) exerts either a dynamic or static force vector which results in a torsion respectively bending moment on all involved joints and bones. Over the whole kinetic chain, all involved joints and potentially affected boney regions need to be stabilized either by using a support, e.g. the cushion of an exercise machine, or by muscle. Both the approach via supporting cushion as well as the active muscle stabilization result in force vectors respectively torsion/bending moments which are balanced out with the force vectors of the agonistic muscles.
- What about the non-affected joints/regions of the body?
This question is kind of a self-supervisory authority. All of the joints/bones outside of the flux of force don’t experience any training stimuli or additional mechanical loading besides their own body weight.
This methodological approach allows the exercise therapist to estimate if a specific exercise involves an affected bone region mechanically and which adaptation of the exercise would reduce the mechanical loading.
We included a respective remark to Gottlob (2020) and his “flux of force principle” in the revised manuscript.
Ad 4) Thank you for this remark. As this is a viewpoint article, it was important to us authors to present our rich clinical and research experience which has been grown over decades. Considering the large number of references for such a short paper, we think that despite the 15 self-citations we were still representing worldwide research well. Nonetheless, we decided to remove five of those own references which we deemed expendable and since after the revision the number of references increased to 55 hope that this self-citation ratio is now acceptable for you.
Reviewer 3 Report
In this study, the authors explored the challenge faced by Austrian experts in cancer rehabilitation in the prescription of exercise to patients suffering from an increased risk of bone fracture (due to metastatic bone disease or multiple myeloma).
I'm glad I could read this article.
General concept comments
The manuscript is relevant for the field and presented in a well-structured manner.
The manuscript is scientifically sound.
Conclusions are consistent with the evidence and arguments presented.
Specific comments
More than 15% of the cited references was published >10 years ago.
There are 15 self-citations involving “Crevenna R”.
I think this is not a consensus paper. Austrian Medical/Scientific Societies were not involved in this study to my knowledge, and authors are not acting as representatives of those societies. So “consensus paper” should not be used as a keyword.
I propose as keywords (MESH) the following: Assessment, Risk; Fractures, Pathologic”; Rehabilitation.
Author Response
Dear Reviewer 2,
Thank you very much for your positive review and your suggestions of how to improve it. As you will see in our point-to-point response, in comments 1 and 2 we reasoned why we chose the references we did, and followed the recommendations of your comments 3 and 4. We hope that you can follow our arguments and that you are satisfied with the revised version of our manuscript.
Specific comments
1. More than 15% of the cited references was published >10 years ago.
Thank you for pointing this out. Because this is a viewpoint article, that to some extent reflects the clinical opinions of the authors, we have presented the history of our department to reflect our decades of expertise. Moreover, we depicted how the specific research area has evolved over time. These things inevitably call for slightly older references and that is why this short viewpoint article has quite a large number of references at 43. If you look at our manuscript from this perspective, you will see that these older publications only served to reflect the historical development of the research field, not to develop our perspective.
2. There are 15 self-citations involving “Crevenna R”.
Thank you for this remark. Professor Crevenna, who is one of the first authors of this manuscript, is the Head of the Department of Physical Medicine, Rehabilitation and Occupational Medicine of the Medical University of Vienna, Austria. He has been one of the most distinguished researchers in the field of cancer rehabilitation in Austria in the last two decades with close to 200 scientific publications in the field of rehabilitation medicine listed in the PubMed. As we specifically wanted to present the Austrian approach to this topic and Professor Crevenna's contribution, role and history in this research area in Austria is significant, it is impossible not to cite his work. Here again, please take into consideration that we referenced 43 publications in a rather short paper, so 28 of the referenced articles do not involve “Crevenna R” and that the current manuscript is a “viewpoint”-article.
3. I think this is not a consensus paper. Austrian Medical/Scientific Societies were not involved in this study to my knowledge, and authors are not acting as representatives of those societies. So “consensus paper” should not be used as a keyword.
Thank you for this remark, you are totally right. We removed the term “consensus paper” from the list of keywords.
4. I propose as keywords (MESH) the following: Assessment, Risk; Fractures, Pathologic”; Rehabilitation.
Thank you for your suggestion. We revised our list of keywords according to your recommendations.
Addition on February 3rd 2023:
Dear Reviewer,
During the review process, also another reviewer raised the issue with the number of self-citations being too high. We therefore allowed ourselves to be convinced to address this issue accordingly. We decided to remove five of our 15 self-citations which we deemed expendable. Since after the revision the number of references increased to 55, we hope that this self-citation ratio (10/55) is now acceptable for you.
Round 2
Reviewer 1 Report
1. The discussion should be rather organized around arguments avoiding simply describing details without providing much meaning. A real discussion should also link the findings of the study to theory and/or literature.
2. English is modest. Therefore, the authors need to improve their writing style. In addition, the whole manuscript needs to be checked by native English speakers.
Author Response
Dear Reviewer,
Thank you for your constructive feedback. Following your recommendations, we revised large parts of the manuscript and hope that this revised version is now to your satisfaction.
Reviewer 2 Report
Authors replied to the comments and made significant progress on the discussion.
Minor comment: Table 1 is missing in the revised manuscript.